# Terahertz Beam Steering Concept Based on a MEMS-Reconfigurable Reflection Grating

**DOI:** 10.3390/s20102874

**Published:** 2020-05-19

**Authors:** Xuan Liu, Lisa Samfaß, Kevin Kolpatzeck, Lars Häring, Jan C. Balzer, Martin Hoffmann, Andreas Czylwik

**Affiliations:** 1Faculty of Engineering, University of Duisburg-Essen (UDE), 47057 Duisburg, Germany; kevin.kolpatzeck@uni-duisburg-essen.de (K.K.); haering@nts.uni-duisburg-essen.de (L.H.); jan.balzer@uni-duisburg-essen.de (J.C.B.); Czylwik@nts.uni-duisburg-essen.de (A.C.); 2Faculty of Electrical Engineering and Information Technology, Ruhr University Bochum (RUB), 44801 Bochum, Germany; lisa.samfass@rub.de (L.S.); martin.hoffmann-mst@rub.de (M.H.)

**Keywords:** terahertz beam steering, blazed grating, MEMS

## Abstract

With an increasing number of applications of terahertz systems in industrial fields and communications, terahertz beamforming and beam steering techniques are required for high-speed, large-area scanning. As a promising means for beam steering, micro-electro-mechanical system (MEMS)-based reflection gratings have been successfully implemented for terahertz beam control. So far, the diffraction grating efficiency is relatively low due to the limited vertical displacement range of the reflectors. In this paper, we propose a design for a reconfigurable MEMS-based reflection grating consisting of multiple subwavelength reflectors which are driven by 5-bit, high-throw electrostatic actuators. We vary the number of the reflectors per grating period and configure the throw of individual reflectors so that the reflection grating is shaped as a blazed grating to steer the terahertz beam with maximum diffraction grating efficiency. Furthermore, we provide a mathematical model for calculating the radiation pattern of the terahertz wave reflected by general reflection gratings consisting of subwavelength reflectors. The calculated and simulated radiation patterns of the designed grating show that we can steer the angle of the terahertz waves in a range of up to ± 56.4${}^{\circ }$ with a maximum sidelobe level of −10 dB at frequencies from 0.3 THz to 1 THz.

## 1. Introduction

In the last few decades, applications of terahertz systems in sensing [1,2], imaging [3,4], and communications [5,6] have drawn great attention. The reasons are that terahertz waves can penetrate many dielectric materials that are opaque for optical waves, allow better spatial resolution, and promise an extremely high data rate compared to the microwaves due to their shorter wavelengths (higher carrier frequencies). The applications mentioned above often require beam control techniques to manipulate the shape of the terahertz beam (beamforming) and direction of the terahertz beam (beam steering). Furthermore, beamforming and beam steering based on phased arrays consisting of two or more sources increase the total radiation power by free-space power combining. It overcomes the low power issue caused by the limited radiation power of a single terahertz source and high atmospheric absorption of terahertz radiation. Terahertz beam steering based on various means, such as phased arrays and electromagnetic metasurfaces, has been comprehensively summarized in [7,8]. Generally, there are two kinds of terahertz phased arrays. The first kind is modulating the phase within the microwave or optical bands that are then up or down-converted to the terahertz range [9,10,11,12,13]. In the second kind, the phase of the terahertz wave is directly shifted or modulated, for example, by using liquid crystals [14,15], or graphene-based terahertz phase shifters [16], metasurfaces [17], leaky wave antennas [18], or micro-electro-mechanical system (MEMS)-based diffraction gratings [19,20]. MEMS-based diffraction gratings are more promising and feasible for terahertz beam steering compared to other means for realizing the terahertz phased arrays mentioned above. It is structurally uncomplicated because it does not require coherent feed networks and phase shifters together with their control units. In addition to that, MEMS allows terahertz beam steering with wide steering ranges at different frequencies by reconfiguring the grating structure. Monnai et al. [20] have successfully demonstrated beam steering at frequencies from 0.15 THz to 0.9 THz using a MEMS-based diffraction grating. However, the diffraction grating efficiency is relatively low due to the limited vertical displacement of the cantilevers [20].

The diffraction grating efficiency can be improved and maximized in a desired diffraction order by a blazed grating. In this paper, we introduce a structure that approximates a reflective sawtooth-like blazed grating. The surface is realized with multiple translating subwavelength MEMS reflectors. Every single MEMS reflector has its own 5-bit actuation system with a total throw of 600 μm, which allows them to move up and down stepwise and independently from each other. By changing the number of reflectors per grating period and the vertical displacement of each reflector, discrete steering angles with high directivity at different frequencies can be achieved. We propose a mathematical model to calculate the far-field radiation pattern of the terahertz wave reflected by the designed grating and compare the calculated radiation patterns with results from electromagnetic (EM) simulations.

The paper is structured as follows. In Section 2, we present the device design of the MEMS reflector array. In Section 3, we provide a mathematical model for a general reflection grating and show far-field radiation patterns of the terahertz wave interacting with various configurations of the MEMS reflection grating structure at different frequencies. At last, in Section 4 we verify the proposed mathematical model by EM simulations of the same reflection grating and comparing the radiation patterns.

## 2. Design of the MEMS Reflector System

### 2.1. Dimensions of a Single MEMS Reflector

In this work, we aim to design a MEMS-based structure that approximates a blazed grating for beam steering at frequencies from 0.3 THz to 1 THz. The structure of a sawtooth-shaped grating with its blaze arrow pointing in negative *x* direction is shown in Figure 1a,b.

When the angle of specular reflection at the grating elements aligns with the angle of diffraction of the grating, a blazed grating is formed. Therefore, at normal incidence the equation
(1)$$\theta_{d}=2\theta_{B}$$
holds for blazed gratings, where $\theta_{d}$ is the diffraction angle and $\theta_{B}$ is the blazed angle. We use the subwavelength MEMS reflectors to approximate the geometry of the blazed grating, as shown in Figure 1c,d. Ideally, the width of a single reflector should be chosen as small as possible for an optimal approximation. However, controlling an excessive number of reflectors simultaneously makes the MEMS system technologically less feasible. Therefore, we chose the width of the single reflector to be just small enough so that the additional unwanted diffraction from the individual reflectors in the targeted frequency range could be neglected. The diffraction grating equation at normal incidence
(2)$$\begin{array}{cc} \sin\left(\theta_{d,m}\right) & =m\cdot \frac{\lambda }{\Lambda }\\ \mathrm{with}\;\;m & =0,\pm 1,\pm 2,\cdots \end{array}$$
where $\theta_{d,m}$ is the *m*th-order diffraction angle, λ is the wavelength in free space, and Λ is the grating period, only has solutions when Λ≥λ. Accordingly, unwanted diffraction from individual reflectors can be neglected when the width of a single reflector is smaller than the operating wavelength. Therefore, for beam steering at frequencies from 0.3 THz to 1 THz, we chose the width of a single reflector ${w=\lambda |}_{f=1\;\mathrm{THz}}=300$
μm.

The height of the blazed grating $H_{\mathrm{blazed}}$ where most of the power is concentrated at the first-order diffraction angle is determined by
(3)$$H_{\mathrm{blazed}}=\Lambda \cdot \tan\left(\frac{\arcsin\left(\frac{\lambda }{\Lambda }\right)}{2}\right)$$
at normal incidence. The required height $H_{\mathrm{blazed}}$ of the reflectors for the blazed gratings at different frequencies calculated from Equation (Equation 3) with Λ being an integer multiple of 300 μm is shown in Figure 2. It can be seen that the required heights converge to λ/2 with the increased grating period Λ, and the total throw $h_{\mathrm{total}}=600\;\mu$m of the reflectors is sufficient for forming the geometries of the blazed gratings. Accordingly, we drive each single reflector with a 5-bit actuation system with a total throw of 600 μm.

### 2.2. Design and Simulation of the MEMS System

The adoption of commonly-known optical MEMS phased arrays based on separately actuated out-of-plane electrostatic actuators to terahertz radiation with wavelengths in the millimeter range is difficult due to the strictly-limited displacement range for out-of-plane actuation. Furthermore, the guiding of every reflector by an individual analog voltage requires complex controlling circuitry, including electronic digital-to-analog converters (DAC) connected to each actuator.

The MEMS-based diffraction grating proposed in this paper consists of an integrated mechanical digital-to-analog converter which allows one to directly control the deflection of the reflectors by parallel digital voltage signals, replacing external electronic DACs. The required large deflection for the adoption of the system to the terahertz range is intended to be realized in-plane on silicon-on-insulator (SOI) substrates which are stacked into piles forming a one-dimensional reflective array, as shown in Figure 3a.

The required microsystem is based on a recently presented digital-to-analog converter depending on a 3-bit system consisting of electrostatic actuators and a mechanical amplifier. In this paper, we present a 5-bit system able to perform the required bi-directional throw of 300 μm (Figure 3b). A modified analogous electronic network model of the mechanical system is shown in Figure 3c. Here, the capacitors are associated with mechanical springs and the actuators can be identified as voltage sources, whereas the voltage corresponds to the displacement in the mechanical domain. The 5-bit system consists of five sliders, each associated with one bit. Electrostatic actuators connected to actuator springs $k_{ai}$ generate the displacement of each slider. Due to high throws and low pull-in voltage [21], electrostatic actuators are very appropriate for terahertz beam steering. The sliders displaced by these high-throw electrostatic actuators are connected by connecting springs $k_{ci}$. Additional guiding springs $k_{si}$ assure a translational displacement of the sliders. On the side of the most significant bit (MSB), the output of the DAC $h_{\mathrm{DAC}}$ is directly connected to a compliant mechanical amplifier [22], which amplifies small input displacements into a large translational output throw. Due to the integration of the mechanical amplifier in the system, high displacements can be reached with low actuation voltage.

For the mechanical simulation of the entire microsystem, a 2-dimensional mechanical finite- element-analysis was carried out using the software COMSOL. Therefore, the actuators associated with the bits were successively displaced according to an imposed binary code at the DAC. Simultaneously, the displacements $h_{i}$ at the bits and at the output of the amplifier were evaluated. The designed and simulated 5-bit system (Figure 4a) consisted of 12 actuators on each side. Half of the actuators realized an upwards displacement (logical 1); the other half realized a downwards displacement (logical 0). For each bit, the guiding springs have a stiffness of 10.1 N/m and the connecting springs of 906.0 N/m. The amplifier has a total stiffness of 334.7 N/m and an amplification ratio of 23. The simulation results (Figure 4b) confirm the concept of a 5-bit, bi-directional, almost linear total displacement of 300 μm of the reflector, when each actuator is deflected by 22.5 μm.

By reducing the width of the single reflector, we can adapt our grating design for beam steering at higher frequencies. To fabricate reflectors with a thickness of 300 μm, the handle layer and the device layer of the SOI substrate are functionally swept and the systems are produced on the (thicker) handle layer. This procedure differs from common MEMS fabrication processes, where the structures are fabricated on the (thinner) device layer. To reduce the widths of the reflectors, either substrates with thinner handle layers can be used or the systems can be fabricated on the device layer. This proceeding reduces the demands on the fabrication, especially on the deep-reactive-ion-etching process (DRIE). Consequently, a reduction of the width of the reflector down to 10 μm and less is possible. With this width, we can scale our grating design for beam steering at frequencies up to 30 THz. Since the mechanical stiffness and the electrostatic force are linearly coupled to the width, a reduction of the width has no impact on the actuation and functionality of the system and its single components. However, a reduction of the width of the reflector increases the demands on the electronic activation of the reflective array.

The first technological trials showed that the concept is feasible when based on SOI MEMS technology. The deviation of the functionality of the fabricated reflective array from the analytically and numerically described reflective array is expected to be done in an appropriate, highly controllable manner. The dimensions of the reflective array depend on two factors; namely, the SOI substrate and the fabrication process. The thickness of the handle layer of the SOI substrate defines the widths *w* of the reflectors. According to our supplier, the thickness of 300 μm varies in a range of ±15 μm, resulting in an inaccuracy of ±5%. The accuracy of the fabrication process defines the deflection of the reflector $h_{\mathrm{total}}$ and of the minimum bitwise displacement of the reflector $h_{\min}$. During fabrication, the single SOI chips pass various processing steps, whereby the accuracy of the lithography and the DRIE process has the main influence on the dimensions of the single components. Initial technological trials showed that the thickness of the components does not exceed a reduction of 8% compared to the mask layout, whereby thicker components have a lower percentage reduction of thickness. An adoption of these results to the final mask design ensures that the fabricated component dimensions suit the simulated systems. In a next step, the actuated reflectors are realized and grouped into rows for an experimental verification.

## 3. Mathematical Model

The simplified structure of a MEMS reflector array-based sawtooth-like reflection grating is shown in Figure 5. The gray area represents the reflective surface of the reflectors. We assume a plane wave that is normally incident on the grating. The phase $\Delta \varphi \left(x,z\right)$ of the wave reflected by the reflectors is spatially modulated. It is determined by the *x* and *z*-coordinates of the reflectors.

We place the reflection grating symmetrically to the *z*-axis, so its reflective surface spans the range
(4)$$-\frac{N_{\mathrm{total}}\cdot w}{2}\leq x\leq \frac{N_{\mathrm{total}}\cdot w}{2}$$
along the *x*-axis, where *w* is the width of a single reflector and $N_{\mathrm{total}}$ is the total number of reflectors constructing the reflection grating. The *z*-coordinate of the reflective surface of one grating element $z_{\mathrm{element}}\left(x\right)$ is the summation of multiple rectangular functions with linearly increasing heights.
(5)$$z_{\mathrm{element}}\left(x\right)=\sum_{n=1}^{N}\mathrm{rect}\left(\frac{x+\frac{w\cdot \left(N_{\mathrm{total}}-1\right)}{2}-\left(n-1\right)\cdot w}{w}\right)\cdot \left(n-1\right)\cdot h,$$
where *N* is the number of reflectors per grating period Λ and *h* is the displacement step between adjacent reflectors. The *z*-coordinate of the reflective surface of the complete grating $z\left(x\right)$ repeats the values of $z_{\mathrm{element}}\left(x\right)$ with the grating period Λ.
(6)$$\begin{array}{cc} z\left(x\right) & =z_{\mathrm{element}}\left(x\right)*\sum_{k=0}^{\left\lfloor \frac{N_{\mathrm{total}}}{N}\right\rfloor }\delta \left(x-k\cdot \Lambda \right)\\ & +\sum_{n=1}^{N_{\mathrm{total}}-\left\lfloor \frac{N_{\mathrm{total}}}{N}\right\rfloor \cdot N}\mathrm{rect}\left(\frac{x-w\cdot \left(\left\lfloor \frac{N_{\mathrm{total}}}{N}\right\rfloor \cdot N-\frac{N_{\mathrm{total}}}{2}\right)-\left(n-1\right)\cdot w}{w}\right)\cdot \left(n-1\right)\cdot h. \end{array}$$

The phase shift of the plane wave along the *x*-axis on the reflective surface of the reflectors can then be expressed as the multiplication of the *z*-coordinate of the reflective surface with the wavenumber in free space
(7)$$\begin{array}{c} \Delta \varphi \left(x\right)=z\left(x\right)\cdot \frac{2\pi }{\lambda }, \end{array}$$
where λ is the wavelength of the plane wave in free-space. Now, we assume that the electric field of the plane wave only has a component in *x* direction and no amplitude variation in the *x* and *z* directions. The reflected component of the electric field $E_{\mathrm{source}}\left(x\right)$ of the plane wave just above the reflective surface of the reflectors can be expressed as
(8)$$E_{\mathrm{source}}\left(x\right)=e_{x}\cdot \int_{-\frac{N_{\mathrm{total}}\cdot w}{2}}^{\frac{N_{\mathrm{total}}\cdot w}{2}}E_{0}\cdot e^{j\Delta \varphi (x)}dx,$$
where $E_{0}$ is a constant amplitude of the electric field of the plane wave and $e_{x}$ is the unit vector in *x* direction. The plane wave on the reflective surface can be modeled as the combination of the electric field from many small isotropic radiators according to the Huygens–Fresnel principle, so that in the far-field region, the electric field $E_{\mathrm{FF}}\left(r,\theta \right)$ of the reflected plane wave at distance *r* in a spherical coordinate system can be calculated from the superposition of the contributions from all isotropic radiators:(9)$$\begin{array}{cc} E_{\mathrm{FF}}\left(r,\theta \right) & =\int_{-\frac{N_{\mathrm{total}}\cdot w}{2}}^{\frac{N_{\mathrm{total}}\cdot w}{2}}\frac{E_{\mathrm{source}}\left(x\right)}{|r\left(x,z\right)|}\cdot \exp\left(-jk\left(x,z\right)\cdot r\left(x,z\right)\right)dx, \end{array}$$(10)$$\begin{array}{cc} \mathrm{where}\;r\left(x,z\right) & =\left(\cos\theta \cdot r-x\right)\cdot e_{x}\;+\;\left(\sin\theta \cdot r-z\left(x\right)\right)\cdot e_{z} \end{array}$$(11)$$\begin{array}{cc} \mathrm{and}\;k\left(x,z\right) & =\frac{2\pi }{\lambda }\cdot e_{x}\;+\;\frac{2\pi }{\lambda }\cdot e_{z}, \end{array}$$
where $e_{z}$ is the unit vector in *z* direction. To maximize the grating efficiency in the first diffraction order, the height difference *h* must be chosen so that the geometry of the designed grating is as close as possible to the geometry of a blazed grating. Therefore, *h* is determined by
(12)$$\begin{array}{cc} h & =\nu \cdot h_{\min},\\ \mathrm{where}\;\nu & =\lfloor \frac{H_{\mathrm{blazed}}}{h_{\min}\cdot \left(N-1\right)}\rfloor , \end{array}$$
where $h_{\min}$ is the minimum bitwise displacement of the reflector. With Equations (Equation 3), (11), and (Equation 12), we can then calculate the radiation pattern with the following specifications according to Section 2:$$\begin{array}{c} w=300\;\mu m,\;h_{\min}=18.75\;\mu m,\;h_{\mathrm{total}}=600\;\mu m,\;E_{0}=1,\;\mathrm{and}\;N_{\mathrm{total}}=80. \end{array}$$

According to Equation (2), the maximum steering angle at one single frequency is limited by the smallest possible grating period Λ=N·w or in our case the smallest possible number of reflectors per grating period *N*. To obtain the smallest possible value of *N*, for which the beam is not only successfully steered in the desired angle but also its directivity is significantly higher compared to the undesired diffraction angles, we calculate the radiation patterns of the beam reflected from the grating with increasing *N* starting with N=1. With N=1, the grating period Λ=w=300μm is smaller than the wavelength at frequencies from 0.3 THz to 1 THz; therefore, no diffraction occurs. With N=2, the grating period Λ=2·300μm=600μm is greater than the wavelength λ=500μm at 0.6 THz, so the grating equation (Equation (2)) has solutions. However, its radiation pattern shown in Figure 6a shows a dominant specular reflection. Inserting N=2 into Equation (Equation 12), we get
(13)$$h\approx H_{\mathrm{blazed}}.$$

Inserting Equation (Equation 3) into Equation (Equation 13), and using the theorems $x\geq \sin\left(x\right)$ and $\tan\left(x\right)\geq x$ for positive *x*, we get
(14)$$h\approx H_{\mathrm{blazed}}\geq \frac{\lambda }{2\cdot \Lambda }\cdot \Lambda =\frac{\lambda }{2}.$$

The displacement step $h\approx H_{\mathrm{blazed}}$ results in a phase shift Δφ≥π. The beam experiences twice the phase shift induced by the displacement step *h*: once when the beam is incident on the two reflectors and again when the beam is reflected from the two reflectors. Therefore, the total phase shift induced by the displacement step *h* is 2Δφ≥2π. For single-frequency operation, the phase shift can be wrapped to 2π. Thus, the wrapped phase shift is a lot smaller than the desired phase shift for a blazed grating $\frac{4\pi \cdot H_{\mathrm{blazed}}}{\lambda }$. Consequently, the structure designed with two reflectors per grating period N=2 has poor grating efficiency. With N=3, the first-order diffraction angles on both sides of the grating normal show strong radiation at 0.6 THz, as shown in Figure 6b. The reason is that the ambiguity of 2π results in the ambiguity of the phase shift distribution induced by the displacement step *h* among the three reflectors. The phase shift distribution {0,π,2π} induced when the beam interacts with the three reflectors can also be interpreted as {2π,π,0}. While the power is efficiently steered to the diffraction angle on the left side of the grating normal with the first phase distribution, the power is efficiently steered to the diffraction angle on the right side of the grating normal with the second phase distribution. With *N* equal to or greater than four, we are able to steer the beam to the desired grating angle with its directivity at least 10 dB higher than the sidelobes, as shown in Figure 6c,d. Hence in our design we choose the smallest number of mirrors per grating period N=4 for frequencies from 0.3 THz to 1 THz so that the grating equation (Equation (2)) has solutions and the 2π ambiguity problem is avoided.

The minimum steering angle is limited by the minimum bitwise displacement $h_{\min}$. When the steering angle gets smaller, the required displacement step *h* decreases. The minimum steering angle occurs once the required displacement step *h* falls below the minimum bitwise displacement $h_{\min}$.

Figure 7 shows the radiation patterns that are calculated at the frequencies 0.3 THz, 0.6 THz, and 0.9 THz. The calculated steering angles match the ones determined from Equation (2), and the directivity of the steered beam is at least 10 dB higher than the specular reflection at $0^{\circ }$.

Figure 8 shows the calculated steering range of the reflection grating at frequencies from 0.3 THz to 1 THz. With increasing frequency, the steering range decreases. By changing the direction of the blaze arrow, the same steering range for the other side of the grating normal can be achieved as well. Besides the main lobe in the desired direction and the specular reflection at 0${}^{\circ }$, the radiation patterns of the reflection gratings show distinct grating lobes. For each achievable steering angle, we determine the corresponding directivity of the main lobe at frequencies different from those of the calculated radiation pattern. The results are depicted in Figure 9. The directivity of the MEMS-based reflection grating increases with increasing frequency as the structure becomes electrically larger. It is nearly constant across the steering range and can be perfectly predicted by the theoretical directivity of an equivalent uniform antenna array. The distinct dips in the directivity occur wherever the level of the grating lobes is particularly high. This effect is quantified by the diffraction grating efficiency shown in Figure 10. Low diffraction grating efficiencies are caused by deviations between the geometry of the designed MEMS-based gratings and the geometry of an ideal blazed grating due to the limited vertical displacement resolution of the MEMS actuators.

## 4. Simulation Results

To verify the mathematical model proposed in Section 3, we conducted EM simulations for different configurations of the MEMS reflection grating structure using the finite-difference time-domain (FDTD) solver Empire XPU. Figure 11 shows the simulation setup. The reflectors (in blue color) with the width w=300μm, length l=5000μm, and thickness d=100μm were modeled as perfect electric conductors. The reflection grating consisting of 80 MEMS reflectors lay in an excitation box (in red color) where a plane wave whose polarization was parallel to *x*-axis was normally incident on the grating. Inside of the excitation box, both the incident field and the field reflected by the grating existed, whereas outside the box only the field reflected from the grating was present. Therefore, we specified a region (in yellow color) right above the excitation box to record the near-field of the reflected wave. The far-field radiation pattern was then calculated using the near-to-far-field transformation. The far-field radiation pattern we observed and obtained was the radiation pattern in the upper part of the xz-plane. In the simulation the perfectly matched layer (PML) absorbing boundary condition was used. We left a space of two wavelengths between the PML boundary and the edges of the excitation box as well as the recording region. We created a mesh for the structure based on the following rules: at least 15 cells per wavelength and at least four cells per object and gap. There are 100 temporal sampling points per period at the highest simulation frequency.

In Section 3, we calculated the radiation patterns of nine different configurations of the MEMS reflection grating structure for the maximum steering angle, the center of the steering range, and the minimum steering angle at the frequencies of 0.3 THz, 0.6 THz, and 0.9 THz, respectively. For an intuitive comparison, we simulated the same nine configurations of the MEMS reflection grating structure and obtained the far-field radiation patterns shown in Figure 12. The calculated and simulated radiation patterns show an agreement not only in the main lobe at the desired steering angle, but also in the side lobes. Furthermore, the simulated side lobe levels for all nine different configurations of the MEMS reflection grating structure are almost identical to the calculated ones.

The recorded near-field during the simulation and the calculated electric field of the plane wave on the reflective surfaces of the reflectors are inherently multiplied by a rectangular function due to the finite size of the reflection grating structure. This corresponds to a far-field pattern that has the shape of a sinc-function. Therefore, there are excessive side lobes around the main lobe which can be seen both in the calculated and simulated radiation patterns. This effect is known from the theory of antenna arrays and aperture antennas [23]. The first side lobes of the sinc-function are approximately 13.46 dB lower than the main lobe. The calculated radiation patterns are in excellent agreement with this number. The side lobe level could be improved at the cost of a reduction in directivity by choosing a suitable non-uniform amplitude distribution for the incident electric field. The grating lobes and side lobes can have different undesired effects. In any application, the grating lobes steal power from the main lobe, thereby reducing the radiated power in the desired direction. This is expressed by the grating efficiency discussed above. In certain sensing applications, such as radar, grating lobes and side lobes are particularly deleterious, as they may reduce the angular selectivity of the sensor system.

The calculated and simulated radiation patterns show an agreement in main lobe and side lobes levels. This verifies our mathematical model as an effective tool for estimation of the far-field radiation pattern of the beam reflected from the MEMS grating.

## 5. Conclusions

In this paper, we have presented the device design of a reconfigurable MEMS-based reflection grating for beam steering at frequencies from 0.3 THz to 1 THz. The device can be relatively easily scaled for beam steering at higher frequencies without technological difficulty. The subwavelength MEMS reflectors that are shaped for different grating structures are deflected in-plane by a 5-bit actuator system. Due to this design, we can achieve a higher vertical displacement of single reflectors which increases the diffraction grating efficiency and control each single reflector stepwise and individually. Furthermore, the deflection of the single reflectors is directly controlled by a parallel digital signal, making electronic DACs obsolete. We have provided a mathematical model for general piston reflector array-based reflection gratings. The MEMS reflectors are configured so that their geometries approximate a blazed grating geometry to achieve high grating efficiency. Using the proposed mathematical model, we have calculated the radiation patterns of the terahertz wave at different frequencies reflected from the designed grating structures. The calculated radiation patterns show that with our designed grating structures, we can steer the terahertz wave up to ± 56.4${}^{\circ }$ with a maximum sidelobe level of −10 dB at frequencies from 0.3 THz to 1 THz. We have conducted EM simulations and the agreement between the calculated and simulated radiation patterns in terms of pattern maximum and side lobe levels shows the validity of our proposed mathematical model. Further work shall be to realize the proposed MEMS-based reflection grating and perform radiation pattern measurements.

## Figures and Tables

**Figure 1 sensors-20-02874-f001:**
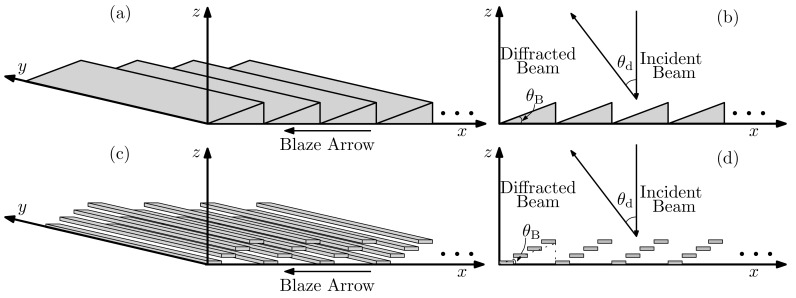
A sawtooth-shaped grating in the (**a**) perspective view and (**b**) side view. An approximate blazed grating using subwavelength MEMS reflectors in the (**c**) perspective view and (**d**) side view.

**Figure 2 sensors-20-02874-f002:**
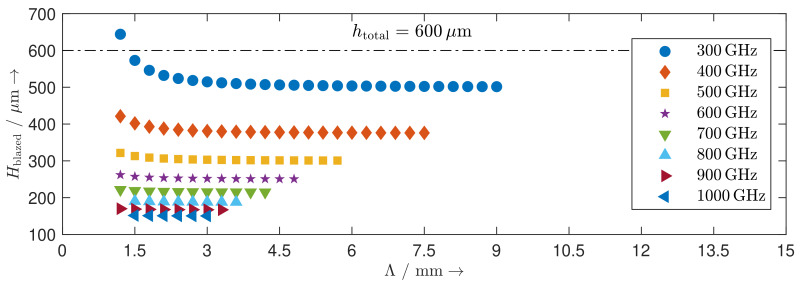
Required heights of the MEMS reflectors for the blazed gratings at different frequencies.

**Figure 3 sensors-20-02874-f003:**
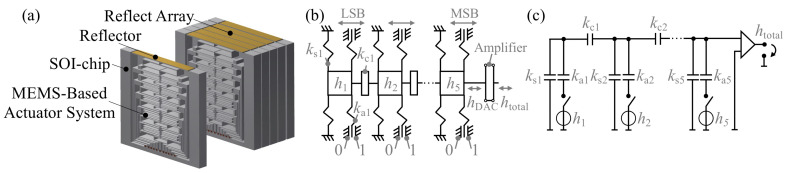
(**a**) Illustration of the reflective array, (**b**) structure of the 5-bit DAC, (**c**) analogous electronic network model.

**Figure 4 sensors-20-02874-f004:**
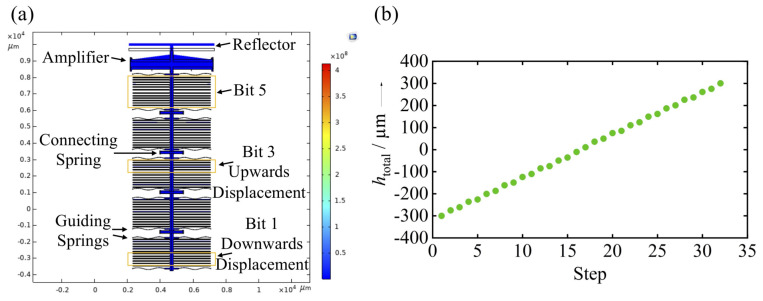
(**a**) Simulated 5-bit system, (**b**) Simulation result: total bi-directional throw of 300 μm performing 32 steps.

**Figure 5 sensors-20-02874-f005:**
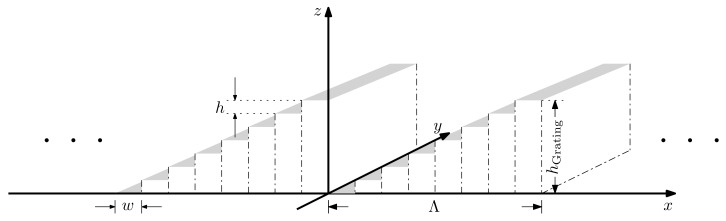
Structure of the MEMS reflection grating.

**Figure 6 sensors-20-02874-f006:**
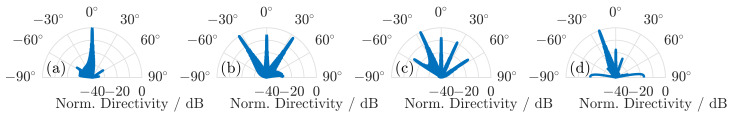
Calculated radiation pattern of a terahertz wave at 0.6 THz reflected by the grating (**a**) with two reflectors per grating period, (**b**) with three reflectors per grating period, (**c**) with four reflectors per grating period, and (**d**) with five reflectors per grating period.

**Figure 7 sensors-20-02874-f007:**
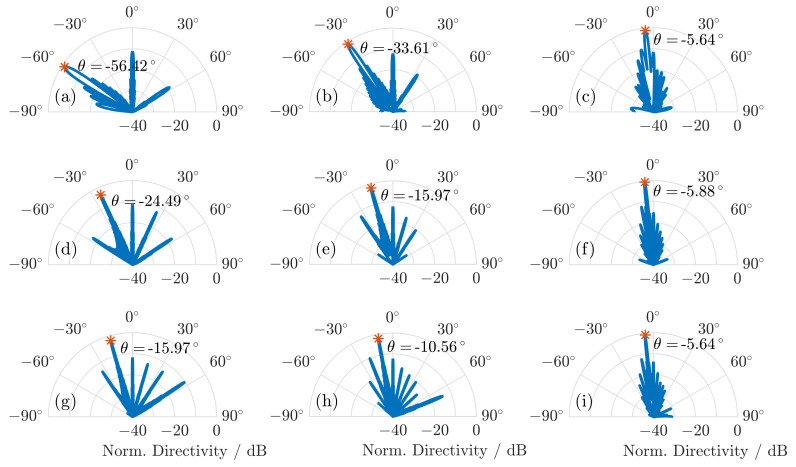
Calculated radiation pattern for (**a**) the maximum steering angle at 0.3 THz, (**b**) the middle of the steering range at 0.3 THz, and (**c**) the minimum steering angle of the steering range at 0.3 THz. Calculated radiation pattern at (**d**) the maximum steering angle at 0.6 THz, (**e**) the middle of the steering range at 0.6 THz, and (**f**) the minimum steering angle of the steering range at 0.6 THz. Calculated radiation pattern at (**g**) the maximum steering angle at 0.9 THz, (**h**) the middle of the steering range at 0.9 THz, and (**i**) the minimum steering angle of the steering range at 0.9 THz.

**Figure 8 sensors-20-02874-f008:**
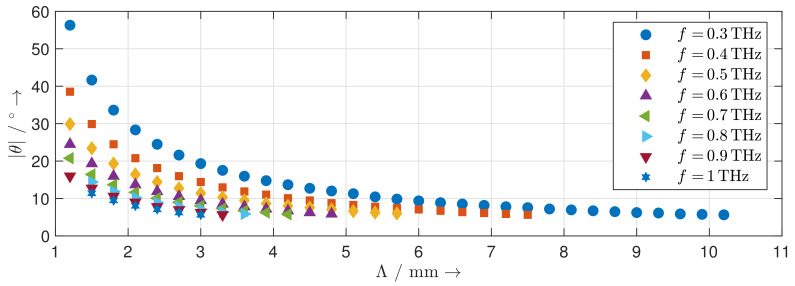
Calculated achievable steering angles of the MEMS reflection grating at different frequencies.

**Figure 9 sensors-20-02874-f009:**
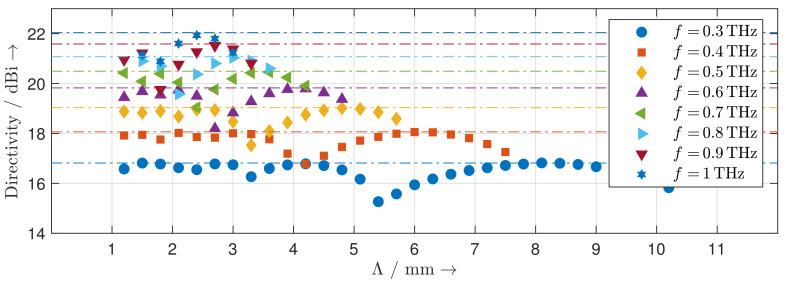
Directivity of the MEMS reflection gratings for different grating periods at different frequencies (solid points) and theoretical directivity of an equivalent uniform linear antenna array (dashed lines).

**Figure 10 sensors-20-02874-f010:**
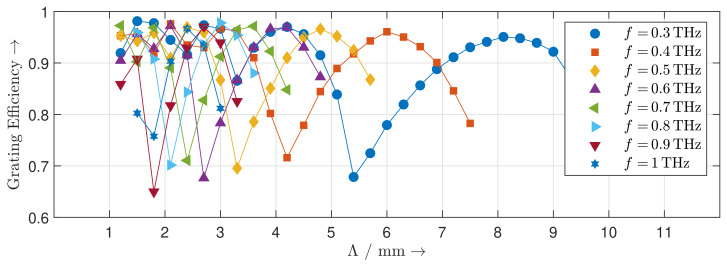
Diffraction grating efficiency of the MEMS reflection gratings for different grating periods at different frequencies.

**Figure 11 sensors-20-02874-f011:**
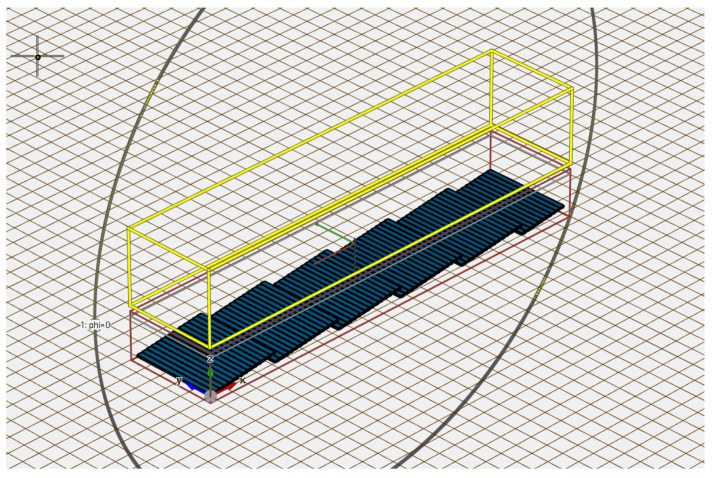
Electromagnetic simulation setup in the FDTD solver.

**Figure 12 sensors-20-02874-f012:**
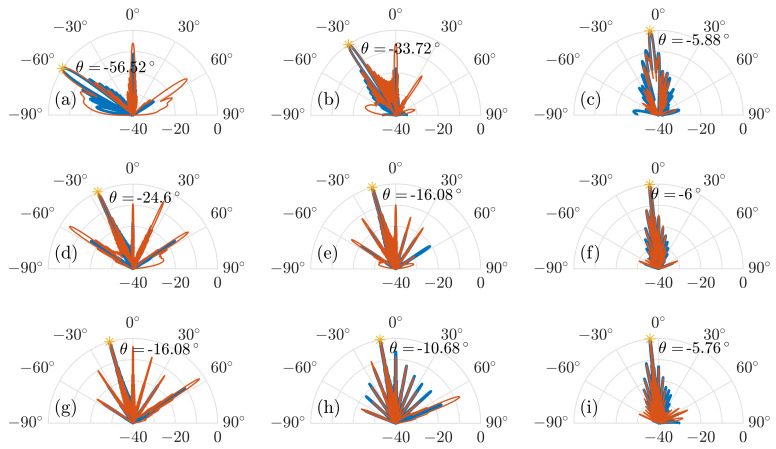
Simulated (in red) and calculated (in blue) radiation pattern at (**a**) the maximum steering angle at 0.3 THz, (**b**) the middle of the steering range at 0.3 THz, and (**c**) the minimum steering angle of the steering range at 0.3 THz. Simulated radiation pattern at (**d**) the maximum steering angle at 0.6 THz, (**e**) the middle of the steering range at 0.6 THz, and (**f**) the minimum steering angle of the steering range at 0.6 THz. Simulated radiation pattern at (**g**) the maximum steering angle at 0.9 THz, (**h**) the middle of the steering range at 0.9 THz, and (**i**) the minimum steering angle of the steering range at 0.9 THz.

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
