# Peer review of "Terahertz Beam Steering Concept Based on a MEMS-Reconfigurable Reflection Grating"

_sensors, 2020, doi:10.3390/s20102874_

Round 1

Reviewer 1 Report

This paper provides theoretical base for author's research in designing a MEMS based reconfigurable reflection grating. It validates their approach in its ability to steer THz beams. Such a grating will be useful for a variety of application spanning data communication, spectroscopy and imaging. I appreciate the two-prong approach to grating modeling - FEM COMSOL and semi-classical Fresnel diffraction theory. To a physicist the fact that results from both models match is expected and validates the correctness of both. This comparison makes this paper stand out from many other modeling papers where only FEM simulations are often presented. The theory of grating modeling is not novel, but it is good to see its use for the design of novel MEMS devices. I am sure many students and scientists will find this paper informative and interesting.

I recommend the paper for publication with minor corrections. 

Line 67. I would replace "diffraction" with "diffraction extrema". The sentence as it stands reads as if there is no diffraction at certain reflector size. 

Line 70. I would replace "diffraction" with "diffraction extrema". For the same reason.

Otherwise the paper is very well written. 

Author Response

We have attached the reply to the reviewer together with the red-lined version of our manuscript which highlights all the changes we have made as the reviewer recommended.

In the red-lined version, it may look like that we removed all the references. But we have only ajusted the format of the references in our paper according to the MDPI reference style guideline.

Reviewer 2 Report

The paper titled “Terahertz Beam Steering Concept Based on a MEMS Reconfigurable Reflection Grating” presents a numerical study of a MEMS based reflection grating. The study is uniquely numerical without any experimental proof. Although the calculations and methodology proposed is comprehensive, the absence of experimental demonstration prevents me from accepting the paper for publication. THz technology is certainly complex and demanding and, hence, the experimental demonstration of technological alternatives is here more necessary than at other regimes where the agreement between simulation and measurement are widely proven. Even in comparatively simpler designs such as single element antennas, the deviations due to manufacturing tolerances usually degrade the performance foreseen in simulations, implying a redesign or recalculation of the structure to adjust the results. In addition, losses at THz are usually underestimated as there are not reliable or universally accepted models for most conductors and dielectrics. Hence, THz demands a stronger experimental effort because, plainly speaking, a simulation is just a first-step approximation to the actual performance of the structure.

Aside from this general comment which is the most important criticism I have on the work, there are other technical aspects that should be considered in more detail:

  1. The gain vs frequency of the structure should be explicitly stated in the manuscript.
  2. There are several examples where the sidelobe level is high. This should be commented in more detail as, in antennas applications, lobes of that level could be unacceptable.

Author Response

(The authors gave the same response as above.)

Reviewer 3 Report

The manuscript presents a theoretical study of a beam steerer working at sub-THz frequencies, based on the actuation of MEMS elements that generate a blazed grating profile. The work is clearly presented and there is very good agreement between theoretical and numerical results. The feasibility and key technological aspects on a possible experimental realization of the proposed device are thoroughly discussed. Overall, it is an interesting proposal in a technologically relevant sector with many applications and future prospect. Publication of the manuscript is recommended, pending the following remarks:

[1] The authors have chosen the width of 300 um of the single element based on which the gratings are designed, which is a reasonable value. This choice dictates the maximum number of elements per wavelength, hence it limits both the maximum diffraction angle and/or efficiency. Please discuss the technological issues related to further reducing the element size in terms of fabrication, actuation and driving conditions. This point is very relevant with respect to the potential scalability of the proposed device to higher frequencies; what would be the maximum operation frequency of the device given current or inherent limitations in MEMS technology?

[2] A minor point: Eq. 2 introduces the grating equation at normal incidence, but Eq. 3 is calculated for m=1; please clarify that the analysis and the results refer to first-order diffraction.

[3] The theoretical approach, as presented in Page 6, is based on the superposition of the reflected field from each element and the calculation far-field radiation pattern, which is elegantly presented. It would be very nice to compare the results with standard scalar diffraction theory for periodic gratings, e.g. the corresponding efficiency of the diffraction orders studied in the results of Fig. 6.

[4] Although it is discussed how the reduction of the number elements/period and hence the poorer discretization of the blazed grating phase profile is expected to lead to lower efficiency, this is not quantified. Figure 8, for instance, can be updated with not only the diffraction angle vs. grating period but also the corresponding efficiencies. This would provide a direct estimation of how the efficiency converges with increasing period and what is the beam steering efficiency of the device at a given angle.

[5] Regarding the FDTD simulations please provide the key simulation parameters: spatial and temporal step and what kind of boundary conditions was employed (PML?). Also, was the total-field/scattered-field formulation was used, as the description of the field excitation/recording points to? Please clarify this point. An additional suggestion, it would be very useful to superpose the results of Fig. 7 on Fig. 10 so that the differences between theory and simulations are clearly visible.

[6] The introduction presents a nice overview of the field of beam steering at THz frequencies. A couple of remarks: although liquid crystals are mentioned as a tunable medium for THz beam steering no references are given. The authors are invited to considered the following very relevant works: doi 10.1109/JSTQE.2019.2956856 and 10.1063/1.5144858. Moreover, vanadium dioxide has also been employed in THz beam steerers, see doi: 10.1038/srep35439.

[7] It is not very well stressed in what aspects does the proposed device outperform existing MEMS-based THz beam-steerers, as those e.g. discussed in the review paper of Ref. [8]. The authors should further highlight the novelty of the proposed solution.

Author Response

(The authors gave the same response as above.)

Round 2

Reviewer 2 Report

The authors have correctly and convincingly addressed my previous comments. The paper can be published in its present form. Nevertheless, an experimental proof of the studied structure would be very desirable in the future

Reviewer 3 Report

I would like to thank the authors for the detailed response they provided to my comments. The manuscript can be now accepted for publication.